# Verification of Saliva Matrix Metalloproteinase-1 as a Strong Diagnostic Marker of Oral Cavity Cancer

**DOI:** 10.3390/cancers12082273

**Published:** 2020-08-13

**Authors:** Ya-Ting Chang, Lichieh Julie Chu, Yen-Chun Liu, Chih-Jou Chen, Shu-Fang Wu, Chien-Hua Chen, Ian Yi-Feng Chang, Jun-Sheng Wang, Tzong-Yuan Wu, Srinivas Dash, Wei-Fan Chiang, Sheng-Fu Chiu, Shin-Bin Gou, Chih-Yen Chien, Kai-Ping Chang, Jau-Song Yu

**Affiliations:** 1Molecular Medicine Research Center, Chang Gung University, Taoyuan 33302, Taiwan; m9301106@gmail.com (Y.-T.C.); julie.chu@mail.cgu.edu.tw (L.J.C.); angela79351990@gmail.com (Y.-C.L.); chihjou415@gmail.com (C.-J.C.); kellyfangwu@gmail.com (S.-F.W.); chienhua0823@gmail.com (C.-H.C.); ianyfchang@mail.cgu.edu.tw (I.Y.-F.C.); dr.kpchang@gmail.com (K.-P.C.); 2Liver Research Center, Chang Gung Memorial Hospital, Linkou 33305, Taiwan; 3National Applied Research Laboratories, Taiwan Instrument Research Institute, Zhubei City, Hsinchu 30261, Taiwan; jason@narlabs.org.tw; 4Department of Bioscience Technology, Chung Yuan Christian University, Taoyuan 32023, Taiwan; tywu@cycu.edu.tw; 5Department of Medical Research, China Medical University Hospital, China Medical University, Taichung 40402, Taiwan; 6Graduate Institute of Biomedical Sciences, College of Medicine, Chang Gung University, Taoyuan 33302, Taiwan; srinivasdash26@gmail.com; 7Department of Oral and Maxillofacial Surgery, Chi-Mei Medical Center, Liouying, Tainan 73657, Taiwan; bigfanfan@yahoo.com.tw (W.-F.C.); jeff60040@hotmail.com (S.-F.C.); ksp_lscamp@hotmail.com (S.-B.G.); 8School of Dentistry, National Yang Ming University, Taipei 11221, Taiwan; 9Department of Otolaryngology, Kaohsiung Chang Gung Memorial Hospital, Kaohsiung 83301, Taiwan; cychien3965@cgmh.org.tw; 10School of Medicine, College of Medicine, Chang Gung University, Taoyuan 33302, Taiwan; 11Department of Otolaryngology-Head and Neck Surgery, Chang Gung Memorial Hospital, Linkou 33305, Taiwan; 12Department of Cell and Molecular Biology, College of Medicine, Chang Gung University, Taoyuan 33302, Taiwan; 13Research Center for Food and Cosmetic Safety, College of Human Ecology, Chang Gung University of Science and Technology, Taoyuan 33303, Taiwan

**Keywords:** oral cancer, saliva, biomarker, MMP-1, ELISA

## Abstract

Oral squamous cell carcinoma (OSCC) accounts for >90% of cases of oral cancer, including cancer at the lip and oral cavity and cancer at the oropharynx. Most OSCCs develop from oral potentially malignant disorders (OPMDs), which consist of heterogeneous lesions with different malignant transformation potentials that make early detection of OSCC a challenge. Using a targeted mass spectrometry-based assay to compare multiple candidate proteins, we previously identified matrix metalloproteinase-1 (MMP-1) as one of the most promising salivary OSCC biomarkers. To explore the clinical utility of MMP-1 in OSCC detection, we developed an in-house, sensitive enzyme-linked immunosorbent assay (ELISA) for measuring MMP-1 content, and tested it on saliva samples from 1160 subjects (313 healthy controls, and 578 OPMD and 269 OSCC patients) collected at two medical centers. Salivary MMP-1 levels measured by our in-house ELISA significantly discriminated OSCC patients from non-cancerous groups. A receiver operating characteristic curve analysis showed that MMP-1 was effective in separating non-cancer groups from patients with OSCCs at the oral cavity. Additionally, salivary MMP-1 levels in oral cavity cancer patients were highly correlated with tumor progression (tumor size, lymph node metastasis, and overall stage). Collectively, our results indicate that salivary MMP-1 is an effective biomarker for OSCC that can be sensitively detected using our newly developed ELISA. The newly developed MMP-1 ELISA may be used as a new adjunctive tool to aid in detecting and monitoring OSCC.

## 1. Introduction

Oral cancer is the sixth-most common cancer worldwide [1,2]. According to World Health Organization statistics, more than 657,000 new cases of oral cancer are reported each year [1,3]. The incidence rate increases year by year, and the total number of cases is expected to reach 856,000 per year by 2035. More than half of these new cases will occur in Asia, and the proportion of cases that end in death is as high as 68% [4]. The average 5-year survival rate for oral cancer patients is about 50% [4]. In Taiwan, oral, oropharyngeal, and hypopharyngeal cancer are the fourth-leading cause of death for all male malignancies, with 7000 newly diagnosed patients and 3000 deaths each year [5]. According to statistics from 2012 to 2016, the survival rates of patients with oral cancer are 79.9%, 71.0%, 56.5%, and 35.6%, for stage I, II, III, and IV, respectively [5]; these data indicate that survival rate can be improved through early detection. Numerous literature reports also point out that the earlier oral cancer is detected, the higher the survival rate [6].

Oral squamous cell carcinoma (OSCC) accounts for over 90% of oral cancers [7]. Most OSCCs are thought to develop from premalignant mucosal lesions associated with an increased risk of malignant transformation (MT) to invasive cancer, termed oral potentially malignant disorders (OPMD) [8]. Conditions including leukoplakia (LE), erythroplakia (EP), erythroleukoplakia, lichen planus (LP), oral lichenoid lesions (OLL), oral submucous fibrosis (OSF), and verrucous hyperplasia (VH) are the most common OPMDs encountered in clinical practice [9,10,11]. Each type of OMPD has a highly variable MT rate and risk of development into invasive carcinoma [8].

Oral mucosal screening is currently an important method for clinical evaluation of OPMD and OSCC. An examination of issue biopsies from the lesion site is the “gold standard” for diagnosis. However, accurately selecting a biopsy site is challenging because of the heterogeneous character of OPMD and OSCC. According to data from World Health Organization (WHO), the sensitivity of oral mucosal screening is 0.5–0.99 and the specificity is 0.64–0.99 [12]. Moreover, biopsy is an invasive technique that requires the compliance of high-risk people, adds to the health cost burden, and is technique-intensive for clinicians; therefore, it may delay the diagnosis of oral cancer [6,13]. It is estimated that over half of patients experience a delay in diagnosis, and more than 50% of patients are in an advanced stage of disease at the time of diagnosis [13]. Therefore, early detection is a key to effective management of oral cancer. Saliva is a biological fluid with features such as ease of sampling, cost-effectiveness, and non-invasiveness that make it a perfect diagnostic specimen, highlighting the importance of developing a method for early detection of OSCC based on validated salivary biomarkers [14,15,16,17,18]. Several well-designed studies recently reported on potential detection methods for oral cancer, suggesting analysis of salivary RNA and proteins [19,20], microRNAs [21,22], metabolites [23,24], glycoprotein [25,26,27,28,29], and the microbiome [30,31,32,33,34,35,36]. However, as yet, no suitable biomarkers are regularly used in clinical practice for the management of oral cancer.

We previously applied a mass spectrometry (MS)-based quantitative approach (multiplex LC-MRM-MS) to simultaneously compare the performance of 49 proteins that are reported to be abnormally present in oral cancer patients as potential salivary biomarkers for oral cancer detection [37]. From this verification study, matrix metalloproteinase-1 (MMP-1) was identified as the most promising candidate [37]. We found differences in MMP-1 levels between saliva samples of OSCC patients and controls up to 83-fold (compared with a fold difference of other proteins of −1.3–5.5), and showed that the area under the curve (AUC) of the receiver operating characteristic (ROC) curve was 0.871 (compared with 0.367–0.870 for other proteins). MMP-1, also known as collagenase-1, is the earliest-identified MMP family member involved in extracellular matrix (ECM) remodeling and is closely associated with metastasis, angiogenesis, and inflammation in tumorigenesis [38]. The inactive pro-form of MMP-1 is secreted via a signal peptide-triggered secretion pathway and becomes activated by removal of the pro-peptide by a serine protease or other MMPs, such as MMP-3 [38,39,40,41,42,43,44]. Active MMP-1 is involved in ECM remodeling through cleavage of multiple types of collagens and several matrix proteins (e.g., fibronectin, gelatin, lamin), and its activity is regulated by binding of tissue inhibitors of metalloproteinase (TIMPs) or autolytic cleavage [39]. Expression of MMP-1 is low in normal resting tissue and is upregulated in physiological contexts such as embryonic development and tissue repair, as well as in pathological processes, including chronic cutaneous ulcers [39,43]. Upregulated expression of MMP-1 has also been reported in several types of cancer [39,45,46,47,48], including oral cancer. Elevated mRNA and protein levels of MMP-1 have been confirmed in both tissue and saliva specimens from OSCC patients [49,50,51,52,53,54]. As the MS-based quantitation method is not suitable for routine quantitative analysis of large numbers of samples in clinical practice, we here produced specific antibodies against MMP-1 and developed an enzyme-linked immunosorbent assay (ELISA). To further verify the effectiveness of MMP-1 as a salivary biomarker for oral cancer detection, we used this in-house-developed ELISA to analyze levels of MMP-1 in 1160 clinical saliva samples from healthy subjects and patients with OPMD or OSCC.

## 2. Materials and Methods

### 2.1. Study Population

Prior to the collection of pre-treatment saliva samples, each subject signed an informed consent form approved by the Institutional Review Board (IRB) of Chi-Mei Medical Center (CMMC, the IRB approval number: 10012-L2, approval date: 08 December 2011) (Tainan, Taiwan) or Kaohsiung Chang Gung Memorial Hospital (CGMH-KH, the IRB approval number: 101-4921B, approval date: 29 January 2013) (Kaohsiung, Taiwan). In this retrospective study, a total of 1160 saliva samples were collected from 313 healthy controls (HC), 578 individuals with OPMD and 269 patients with OSCC (242 at the lip and oral cavity and 27 at the oropharynx) at CMMC and CGMH-KH Medical Center from 2011 to 2016 (Table 1). Two hundred and eighty-four healthy controls were recruited from the subjects enrolled in the Taiwan’s Oral Cancer Screening Program. The other 29 healthy controls, 578 OPMD and 269 OSCC patients were recruited from the subjects who visited the Department of Oral and Maxillofacial Surgery at CMMC or Department of Otolaryngology at CGMH-KH for oral health exam and/or treatment of OPMD/OSCC. Subjects who were above 20 years old, having behaviors of smoking, and/or betel nut chewing, with oral lesions (OPMD and OSCC groups), and willing to give informed consent (all proper patients we met in the clinic were invited) were enrolled in this study. Subjects having personal history of other cancers or severe diseases were excluded from this study. The final diagnosis of OPMD was made by oral and maxillofacial surgeons or otolaryngologists according to the clinical features or pathological findings. Diagnoses of OSCC were confirmed by biopsy, and patients underwent routine checkups according to a standard protocol. The pathological and nodal stages of all tumors were established as described in the American Joint Committee on Cancer (AJCC) Cancer Staging Manual Eighth Edition (2017). It is noted that because the OSCC subjects enrolled in this study were recruited between 2011 and 2016, information about the depth of invasion (DOI) in the T stage and the extranodal extension (ENE) of the node in the N stage, the two major criteria newly launched in the AJCC 8th edition, were not available for those enrolled cases. Thus, although re-evaluation of the recruited OSCC cases should have been done, it could not be undertaken due to the lack of DOI and ENE records. OSCC cases were classified according to the ICD-O-3 code. Cases of OPMD were classified according to the risk of malignant transformation, as previously described [37], and grouped as low-risk OPMD I and high-risk OPMD II. Detailed information on each case, including the time (year) and site of sample collection, gender, age, individual betel nut chewing or smoking habits, saliva appearance, and other clinical information were recorded and are summarized in Appendix A.

### 2.2. Saliva Collection

Saliva specimens were collected and processed as previously described [37]. In brief, each unstimulated whole saliva sample was collected during oral mucosal examination. Donors were food and water fasted and had refrained from smoking and use of oral hygiene products for at least 1 h prior to collection. Before saliva collection, donors were asked to rinse their mouth with clean water. Passive-drooled saliva samples from donors were collected into 15 mL sterile centrifuge tubes, and the obtained saliva samples were centrifuged at 3000× *g* for 15 min at 4 °C. The supernatant was treated with a protease inhibitor cocktail (Sigma, St. Louis, MO, USA), and aliquots were stored at −80 °C.

### 2.3. Development of MMP-1 ELISAs

MMP-1 content in saliva specimens was examined using two in-house-developed ELISAs: pair A-ELISA and pair B-ELISA. C-terminally His-tagged full-length human MMP-1 recombinant protein was produced in the insect sf21 cell line using a baculovirus expression system and purified to near homogeneity, as previously described [55]. Purified MMP-1 recombinant protein was used as an immunogen to generate anti-human MMP-1 mouse monoclonal antibodies (mAbs), and as a protein standard in subsequent ELISAs. Two mAbs were used as capture Abs, and another two were conjugated with horseradish peroxidase (HRP) and used as detection Abs. Clone 31-34 and clone 73-1 were used as capture and detection Ab pairs in pair A-ELISA, and clone 6-2 and clone 20-4, respectively, were used in pair B-ELISA. Detailed information and data on functional evaluation of recombinant human MMP-1 protein and human MMP-1-specific mouse monoclonal Abs are presented in Appendix A.

### 2.4. Measurement of Salivary MMP-1

For the determination of salivary MMP-1 content, each saliva specimen was analyzed simultaneously using the two in-house-developed ELISA sets. The 1160 saliva specimens were blind-tested in a total of 32 96-well microplates. Each assay plate contained a standard curve series (in duplicate), three controls, and 76 saliva samples. The standard curve was prepared by making 3-fold serial dilutions of a 20,000 pg/mL MMP-1 protein standard down to 27.4 pg/mL; 0 pg/mL was used as a blank. Three controls were prepared by spiking saliva specimens from healthy donors with MMP-1 (500, 1000, and 5000 pg/mL) after first testing undiluted saliva samples. Initial data are summarized in Appendix A. Samples generating optical density (O.D.) values near or higher than the highest standard point were re-tested at both 2.5- and 5-fold dilutions (Appendix A). ELISAs were performed using the following general protocol: (1) 96-well microplates (8-well/strip × 12 strips) were pre-coated with the capture Ab, blocked, dried, and stored at 4 °C until used; (2) microplates were prepared by adding Blank, Standards, Controls or samples to wells, then incubated for 1 h at room temperature (RT) on shaker; (3) after washing wells four times, detection Ab was added to the wells, and plates were incubate for 40 min at RT on a shaker; (4) after washing again, 3,3′,5,5′-tetramethylbenzidine solution was added to wells and plates were incubated for 20 min at RT on a shaker; (5) stop solution was added to wells and mixed thoroughly by incubating plates for 2 min on a shaker; (6) the O.D. value of each well at 450 nm was determined within 30 min with a microplate reader, using 540 nm or 570 nm as the reference wavelength; (7) the MMP-1 concentration in each sample was calculated by reference to the standard curve.

### 2.5. Statistical Analysis

Levels of salivary MMP-1 in each study group were represented by box plots, where the middle line of the box corresponds to the median; box edges denote the 25th (lower edge) and 75th (upper edge) percentiles; whisker caps indicate the 10th (lower) and 90th (upper) percentiles; and symbols outside caps indicate 5th (lower) and 95th (upper) percentiles. All data were analyzed using R software, all tests were two-sided, and a *p* value < 0.05 was considered statistically significant. The distribution of sample characteristics between study groups was analyzed using a chi-squared test for categorical variables and a Kruskal–Wallis test for continuous variables. Differences in salivary MMP-1 levels between groups were tested using either a Wilcoxon test (between two groups) or Kruskal–Wallis test (among more than two groups); Dunn’s test was used in post-hoc analyses. Receiver operator characteristic (ROC) curves were constructed by plotting sensitivity versus 1-specificity. The confidence interval (CI) of the area under ROC curves (AUCs) was analyzed using the pROC package [56], which employs DeLong’s method to estimate the standard error of AUCs. The correlation of salivary MMP-1 levels between the two ELISAs was calculated using Pearson correlation coefficient.

## 3. Results

### 3.1. High Salivary MMP-1 Levels in OSCC Patients

Using the newly developed ELISAs, we examined MMP-1 levels in a total of 1160 saliva samples collected from 313 healthy controls (HC), 305 OPMD I patients, 273 OPMD II patients, and 269 OSCC patients. As shown in Figure 1A,B, MMP-1 levels in saliva from OSCC patients were significantly higher than those in non-cancerous groups (HC, OPMD I and OPMD II) (*p* < 0.001), exhibiting fold-differences in median concentration of 22.1–60.9 and 15.3–35 based on pair-A ELISA and pair-B ELISA data, respectively. Additionally, ROC curve analyses showed good efficacy of salivary MMP-1 level in discriminating OSCC from non-cancerous groups (HC, OPMD, HC and OPMD) (Figure 1C,D), with AUCs of 0.871–0.898 and 0.865–0.885 based on pair-A ELISA and pair-B ELISA data, respectively. Using Youden’s J statistic to select the best cut-off value (maximum of Youden’s index) for comparisons between non-cancerous groups (HC and OPMD) and OSCC, we found that the cut-off value, sensitivity, specificity, and accuracy of pair-A ELISA were 256.5 pg/mL, 76.58%, 86.76%, and 84.40%, respectively (Table 2), whereas the corresponding values for pair-B ELISA were 213.4 pg/mL, 79.93%, 83.5%, and 82.67%. Moreover, a consideration of differences in the distribution of sample characteristics between study groups (Table 1) led us to perform a characteristics adjustment analysis, which revealed significant significances among the four study groups (Appendix A). The distribution of salivary MMP-1 in each study group was similar between the two ELISA pairs (Appendix A). Since MMP-1 levels determined by the two ELISA pairs were close to each other in most cases and were highly correlated (Pearson’s *r* = 0.957, *p* < 0.001) (Figure 1E), subsequent analyses were performed using only data obtained by pair A-ELISA.

### 3.2. Differential MMP-1 Levels in OSCCs Located at Different Sites

In Taiwan, cheek mucosa and tongue represent the two major sites for occurrence of OSCC [5]. To gain insight into relationships between salivary MMP-1 levels and OSCCs at different sites, we classified the OSCC group according to site based on the ICD-O-3 code (Figure 2A) and presented the salivary MMP-1 level of each case by scatter plot for statistical analysis (Figure 2B). As shown in Figure 2B, distribution of MMP-1 levels in OSCC patients was more diverse than that in the non-cancerous group. The median MMP-1 concentrations were 31.9 pg/mL for the non-cancerous group, 288.1–2077.7 pg/mL for patients with lesions located at the oral cavity, and 0–554.3 pg/mL for patients with lesions at the oropharynx, respectively. Due to limited sample size, only subgroups with case numbers greater than 25 were subjected to statistical analysis. MMP-1 levels in saliva from patients with lesions located at other and unspecified parts of tongue (C02), gum (C03), cheek mucosa (C06.0), and multiple sites at the oral cavity were significantly higher than those in saliva from non-cancerous groups (HC and OPMD) (*p* < 0.001) (Figure 2B). The AUCs of the four subgroups versus non-cancerous groups were 0.896, 0.936, 0.900, and 0.877, respectively (Table 2). The sensitivity of the four subgroups was between 71.38% and 84.38%, with specificity of 83.28–92.59% and accuracy of 83.35–92.02% at the best cut-off value of 190.5–489.2 pg/mL (Table 2). Moreover, an additional ROC curve analysis of overall OSCCs at the oral cavity and at oropharynx versus non-cancerous groups showed AUC values of 0.899 and 0.738, respectively. The sensitivity of the two subgroups was 79.32% and 70.37%, with specificity of 86.76% and 74.64%, and accuracy of 85.20% and 74.51% at the best cut-off value of 256.6 and 122.5 pg/mL, respectively (Table 2). These data indicate that salivary MMP-1 levels can be used to effectively discriminate OSCCs occurring at the oral cavity including cheek mucosa, gum, tongue (excluding base of tongue), and multiple sites at the oral cavity from non-cancerous groups (HC and OPMD).

### 3.3. Elevated Salivary MMP-1 Levels in OPMD II Patients

During our comparison of salivary MMP-1 levels between non-cancerous groups and OSCCs, we noticed that, in non-cancerous groups, salivary MMP-1 levels in OPMD II patients were significantly higher than those in OPMD I and HC subjects (*p* < 0.001) (Figure 1A) and showed differences in median MMP-1 concentration of about 2.3–2.8-fold (20.9, 24.8, and 57.7 pg/mL in HC, OPMD I, and OPMD II, respectively). As no significant difference was found between OPMD I and HC groups, the box-plotted distribution of salivary MMP-1 levels in the two groups were almost the same (Figure 1A). As shown in Figure 3B, a further analysis revealed that the box-plotted distribution of salivary MMP-1 levels in patients from different types of OPMD II (diagnostic types 1–5, accounting for 96.0% of OPMD II) was more diverse than that in the OPMD I group (diagnostic types 1 and 2, accounting for 97.7% of OPMD I) (Figure 3A). The distribution of MMP-1 levels in 287 patients with leukoplakia (OPMD I, type 1) was similar to that in the HC group, with the same median concentration of 20.9 pg/mL. Notably, there were relatively high levels (4.1–4.3-fold increase) of salivary MMP-1 in OPMD II patients diagnosed with high-grade oral submucous fibrosis (OSF) (type 2, 84.9 pg/mL) or verrucous hyperplasia (VH) (type 3, 90.0 pg/mL) compared with patients diagnosed with erythroleukoplakia or speckle leukoplakia (type 1, 56.9 pg/mL), non-homogeneous leukoplakia (type 4, 44.7 pg/mL), or OSF accompanied by precancerous lesions (type 5, 46.1 pg/mL). These data indicate that the levels of salivary MMP-1 were slightly elevated in OPMD II patients, and that there was a significant difference between the levels of salivary MMP-1 in OPMD II type 3 and OPMD I type 1 (*p* < 0.05), and between OPMD II type 3 and HC (*p* < 0.01). Although levels of salivary MMP-1 were relatively higher in subgroups of OPMD II compared with HC and OPMD I groups, the levels of salivary MMP-1 in the OSCC group were ~22.1-fold higher than those in the OPMD II group, with medium MMP-1 concentrations of 1273.2 and 57.7 pg/mL, respectively.

### 3.4. The Efficacy of Salivary MMP-1 as a Biomarker for Detecting Early-Stage OSCC at the Oral Cavity

As MMP-1 was prominently elevated in oral cavity cancer patients compared with non-cancerous groups, we further analyzed salivary MMP-1 levels in patients according to overall stage. As shown in Figure 4A, MMP-1 levels in patients were elevated in stages I to IV (*p* < 0.001) and varied significantly between stage IV and stage I patients (*p* < 0.05), indicating an increase in MMP-1 levels with OSCC progression. Moreover, salivary MMP-1 levels in stages I, II, III, and IV oral cavity cancer patients were higher than those in non-cancerous groups (HC and OPMD), with AUCs of 0.839, 0.908, 0.899, and 0.946, respectively (Figure 4B). The sensitivity, specificity, and accuracy with overall stage I–IV was 79.49–88.64%, 71.60–90.80%, and 72.50–90.32% at the best cut-off value (determined by the Youden’s index) from 97.5 to 397.2 pg/mL, respectively (Table 2). When the assay is going to be applied in clinical practice for detecting OSCC among the at-risk populations, it would be more practical to set a fixed cut-off value. Two cut-off values, 200 pg/mL (accuracy >80% in all comparison groups) and 100 pg/mL (with sensitivity >80% in detecting stage I OSCC) were selected to show their sensitivity, specificity, and accuracy in different comparison groups of this study (Appendix A). The median MMP-1 concentration in saliva from stages I, II, III, and IV OSCC patients was 382.5, 1324.4, 2912.6, and 2775.4 pg/mL, respectively, and the median fold-difference between the corresponding stage and OPMD II (median, 57.7 pg/mL) was 6.6-, 23.0-, 50.5-, and 48.1-fold, respectively. These observations suggest that salivary MMP-1 might be a useful biomarker for monitoring malignant transformation from OPMD to OSCC.

### 3.5. Higher Levels of MMP-1 in Oral Cavity Cancer Patients with Advanced Clinical Stages and Grades

As MMP-1 was elevated according to the overall stage, we next analyzed salivary MMP-1 levels in OSCC patients with lesions at the oral cavity according to factors associated with advanced cancers, including tumor size (pT status), infiltration of nearby lymph nodes (pN status), and differentiation of cancer cells (grade). As shown in Figure 4C–E, levels of salivary MMP-1 were significantly increased with increasing pT status (T0–T4), pN status (N0–N3), and grade (G0–G3). The median concentration of salivary MMP-1 in patients with T0-1 (tumor diameter < 2 cm), T2 (2–4 cm), T3 (>4 cm), and T4 (tumor infiltrated to nearby tissue) was 429.6, 1975.9, 4620.1, and 2294.5 pg/mL, respectively. A comparison showed that levels of salivary MMP-1 were significantly different between patients with T0-1 and T2 (*p* < 0.01), T0-1 and T3 (*p* < 0.001), and T0-1 and T4 (*p* < 0.001) (Figure 4C). In addition, salivary MMP-1 levels in patients with one infiltrated lymph node (lymph node > 3 cm) or multiple infiltrated lymph nodes (N2-3) were higher than those in patients without lymph node infiltration (N0) (*p* < 0.001, Figure 4D); the median MMP-1 concentration in patients with N0, N1, and N2-3 was 1102.7, 1415.4, and 2977.3 pg/mL, respectively. These results coincide with the upregulated expression levels of MMP-1 in cancerous tissue and the functional role of MMP-1 in extracellular matrix (ECM) remodeling associated with cancer cell metastasis in tumorigenesis. Our data further indicate that salivary MMP-1 levels in patients with undifferentiated and well-differentiated (G0-1), moderately differentiated (G2), and poorly differentiated (G3) cancers were 1150.3, 1273.2, and 5206.3 pg/mL, respectively. These results clearly indicate that salivary MMP-1 levels in patients with poorly or moderately differentiated cancer are higher than those in patients with undifferentiated or well-differentiated cancers, with a median fold-difference of 4.5- or 1.4-fold (*p* < 0.05) (Figure 4E). The notable increase in salivary MMP-1 levels in oral cavity cancer patients from G0-1 to G3 and N0 to N2-3 suggests that salivary MMP-1 might be a useful indicator of OSCC prognosis.

## 4. Discussion

With its characteristics of late diagnosis, high mortality rates, and morbidity, oral cancer, among the more prevalent cancers worldwide, represents a global public health problem. Based on the presumption that early diagnosis and timely therapy can prolong patient survival, our team has worked for years on the development of biomarkers for early diagnosis and prognostic assessment of oral cancer patients [37,55,57,58,59,60]. Accumulating evidence from different groups suggests that salivary levels of MMPs, including MMP-1, MMP-2, MMP-3, MMP-7, MMP-8, MMP-9, MMP-10, MMP-12, and MMP-13, have potential as diagnostic markers of OSCC [52,61,62,63,64,65,66,67,68].

Our previous study of a cohort of 460 samples based on an MS-based quantification method found that, among 28 published biomarker candidates, salivary MMP-1, MMP-3, and MMP-9 protein were all elevated in the OSCC population, and further verified salivary MMP-1 protein as the most promising biomarker for discriminating OSCC from the high-risk healthy population [37]. However, very few of these markers have been verified in a larger number of case and control samples. Consistent with the results of our previous results obtained using a targeted MS-based assay, we found using our newly developed ELISA that salivary MMP-1 is a promising biomarker for discriminating OSCC patients from at-risk non-OSCC subjects.

In addition, as overall OSCC stage progressed from stage I to stage IV, salivary MMP-1 levels in OSCC patients increased significantly, suggesting that MMP-1 could be used to monitor OSCC progression. In recent years, it has been shown that MMP-1 protein is an unfavorable prognostic marker for a number of cancers, including salivary gland cancer [69], nasopharyngeal carcinoma [70], gastric cancer [71], thyroid follicular carcinomas [72], esophageal squamous cell carcinoma [73], and OSCC [74]. According to our analysis, salivary MMP-1 levels in OSCC patients with poorly differentiated cancer (G3) were higher than those in patients with undifferentiated or well-differentiated (G0-1) cancers. Moreover, we found that salivary MMP-1 levels in OSCC patients with one infiltrated lymph node (lymph node > 3 cm) or multiple infiltrated lymph nodes (N2-3) were higher than those in patients without lymph node infiltration (N0). These data demonstrate that levels of salivary MMP-1 are correlated with cancer stage and lymph node infiltration, and strongly suggest that salivary MMP-1 is also an indicator of poor prognosis for patients with OSCC.

In addition to MMP-1, MMP-9 has also been suggested in several studies as a potential biomarker for OSCC [63,64,65,66,67]. Recently, Smriti et al. used a commercial ELISA kit to evaluate salivary MMP-9 levels in 88 cases, including OSCC (*n* = 24), OPMD (*n* = 20), tobacco users (*n* = 22), and healthy controls (*n* = 22), collected from a hospital in India [66]. Their data showed that the mean ratio of salivary MMP-9 levels in OSCC or OPMD relative to control groups was 1.81 and 1.58, respectively. Moreover, patients in the poorly differentiated OSCC group had significantly higher mean saliva MMP-9 levels than those in moderate or well-differentiated OSCC groups. These results indicate that salivary MMP-9 has predictive value for the diagnosis of OSCC and OPMD patients in India [66]. A similar trend was noted in our previous study [37], where we found that the mean ratio of salivary MMP-9 levels in OSCC to control cases in Taiwan was 3.2 [37]. In addition, we also found a significant difference in salivary MMP-1 levels between OSCC and control cases, with a mean ratio of 83.0 [37]. Notably, among the OSCC patient population in India, males accounted for 58% of all patients (male:female = 14:10); however, in our previous and present studies, males accounted for 98.5% (male:female = 129:2) [37] and 97.6% (male:female = 279:7), respectively, of the OSCC patient population in Taiwan. These data reflect the fact that Taiwanese men have the highest incidence of oral cancer in the world [5], and also suggest effects of genetic and lifestyle factors on biomarker variation.

The current gold standard for diagnosing oral cancer and OPMD is a tissue biopsy [64,75]. In 2011, a product that qualitatively detected the presence of human CD44 protein (Vigilant Biosciences, Inc., Austin, TX, USA) for aiding in oral cancer diagnosis was announced. CD44 has been the most frequently reported potential biomarker for OSCC [76,77,78,79]. To compare the efficacy of salivary MMP-1 and CD44 as biomarker for OSCC diagnosis, we also measured and analyzed levels of salivary MMP-1 and CD44 in the same cohort of saliva samples in our previous study [37]. Our results based on tests of saliva samples from 131 oral cancer patients and 199 control cases showed that MMP-1 had a sensitivity of 69.5% and specificity of 95.0%, whereas CD44 had a sensitivity of 76.3% and specificity of 57.8%. ROC curve analyses showed good efficacy of both salivary MMP-1 and CD44 in discriminating OSCC from healthy controls, with AUCs of 0.871 and 0.726, respectively [37]. Taken together, these data support the conclusion that disease biomarkers perform differently among different ethnic groups [80]. They further suggest that salivary MMP-1 is a useful biomarker that is as good or even better than CD44 for OSCC diagnosis in the Taiwanese population.

As mentioned, upregulated expression of MMP-1 in tissue, serum, and/or urine specimens has been reported in several other cancer types [45,46,47,48,70,71,72,73]. However, to our best knowledge, no studies have been carried out to examine the salivary MMP-1 levels in cancer patients other than oral cancer. On the other hand, there were few studies reporting the increase of salivary MMP-1 levels in subjects with periodontal disease, a kind of inflammatory oral disease, as well as the alterations of salivary MMP-1 levels in subjects with periodontal disease who underwent scaling and root planing and/or ozonotherapy [81,82]. As we did not collect the information regarding the periodontitis condition of each enrolled subject, periodontal disease is a confounding variable that was not taken into account in the present study, which represents an important limitation of the study.

## 5. Conclusions

In conclusion, to the best of our knowledge, this is the first report to evaluate salivary MMP-1 levels and demonstrate good diagnostic value for early OSCC diagnosis in a large cohort. In the present study, we suggest that salivary MMP-1 is a good biomarker for (i) early-stage OSCC screening of high-risk populations; (ii) monitoring OSCC progression or disease recurrence; (iii) monitoring the status of neck lymph node metastasis in OSCC. Our results provide the basis for applying salivary MMP-1 content for early-stage screening, predicting transformations, and follow-up monitoring. It should be noted that these are potential future applications, which need to be properly tested with further studies. In the future, confounding variables such as the periodontal disease will be analyzed; moreover, the benefit of applying salivary MMP-1 assay for OSCC screening and monitoring should be evaluated in comparison with a white-light inspection performed by a trained clinician (for screening) and actual gold standards, such as MRI, CT, and biopsies (for monitoring). Given that Taiwan has the world’s highest oral cancer crude incidence rate, it is imperative to develop a good tool for early OSCC diagnosis in this population. Our next step is to develop immunoassay test strips for qualitative detection of salivary MMP-1 that can be used for point-of-care testing [83]. Clearly, a prospective study in a larger cohort comprising salivary samples from patients with different OSCC stages will still be required to further validate the utility of MMP-1 for future clinical application in OSCC therapy.

## Figures and Tables

**Figure 1 cancers-12-02273-f001:**
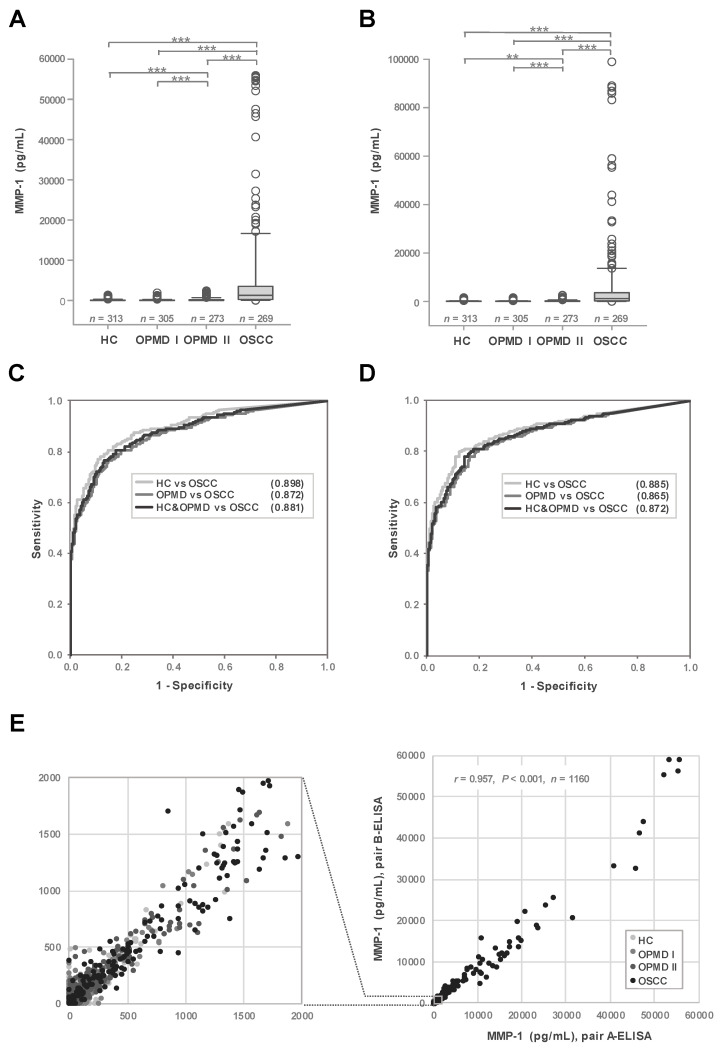
Matrix metalloproteinase-1 (MMP-1) levels in saliva from 1160 clinical specimens. (**A**,**B**) Salivary levels of MMP-1 in healthy controls, oral potentially malignant disorders (OPMD) I, OPMD II, and oral squamous cell carcinoma (OSCC) patients were determined using two pairs of enzyme-linked immunosorbent assays (ELISAs), pair A (**A**) and B (**B**), as described in Section 2. ** *p* < 0.01. *** *p* < 0.001. (**C**,**D**) Receiver operating characteristic (ROC) curve showing the diagnostic efficacy of salivary MMP-1, determined using pair-A ELISA (**C**) and pair-B ELISA (**D**) for OSCC versus non-cancerous groups (healthy controls (HC) and OPMD). (**E**) Correlation between levels of salivary MMP-1 in HC, OPMD I, OPMD II, and OSCC patients, determined using pair A- and pair B-ELISA, as described in Section 2 (Pearson correlation, *r* = 0.957, *p* < 0.001). The boxed image in the right panel is enlarged and shown on the left.

**Figure 2 cancers-12-02273-f002:**
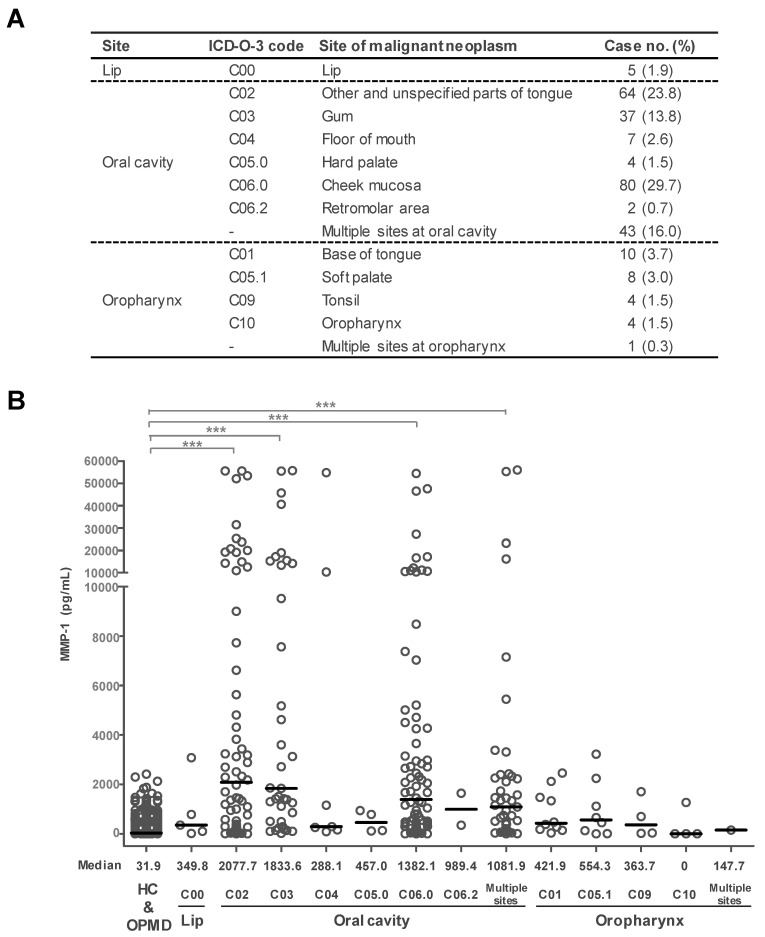
Differential expression levels of salivary MMP-1 in OSCC and non-cancerous groups. (**A**) Clinical information and subgroups of OSCC (*n* = 269) in this study. (**B**) Associations between salivary MMP-1 content in HC, OPMD, and OSCC with different tumor sites. *** *p* < 0.001. The median level of salivary MMP-1 (pg/mL) in each subgroup is shown for comparison.

**Figure 3 cancers-12-02273-f003:**
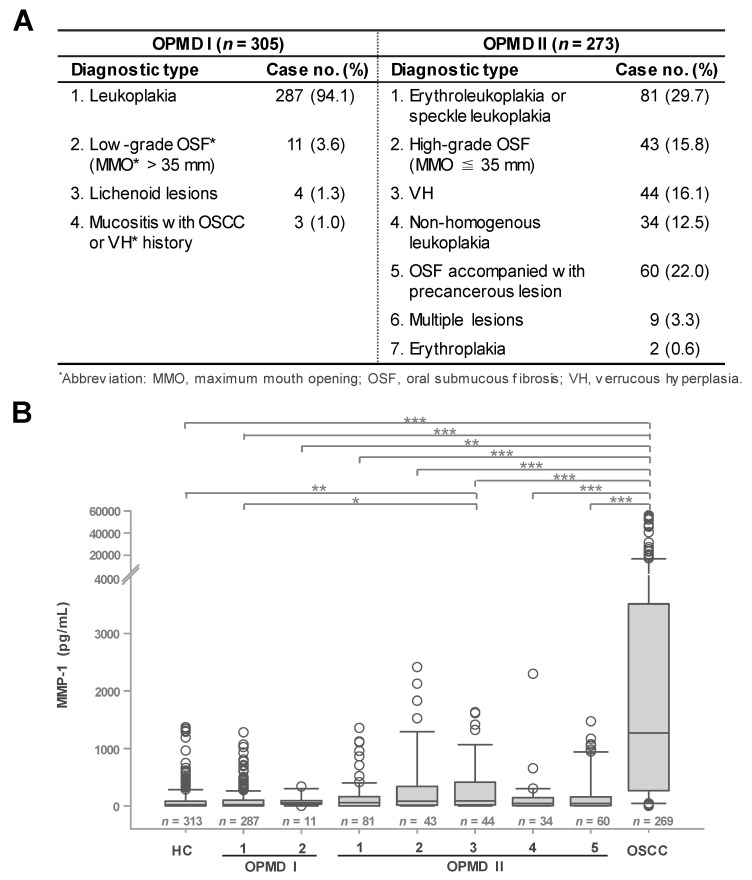
Differential expression levels of salivary MMP-1 in OPMD I and OPMD II. (**A**) Diagnostic types of OPMD I and OPMD II cases in this study. (**B**) Associations between salivary MMP-1 content in HC, different types (case no. > 10) of non-cancerous OPMD I and OPMD II, and OSCC. * *p* < 0.05, ** *p* < 0.01. *** *p* < 0.001.

**Figure 4 cancers-12-02273-f004:**
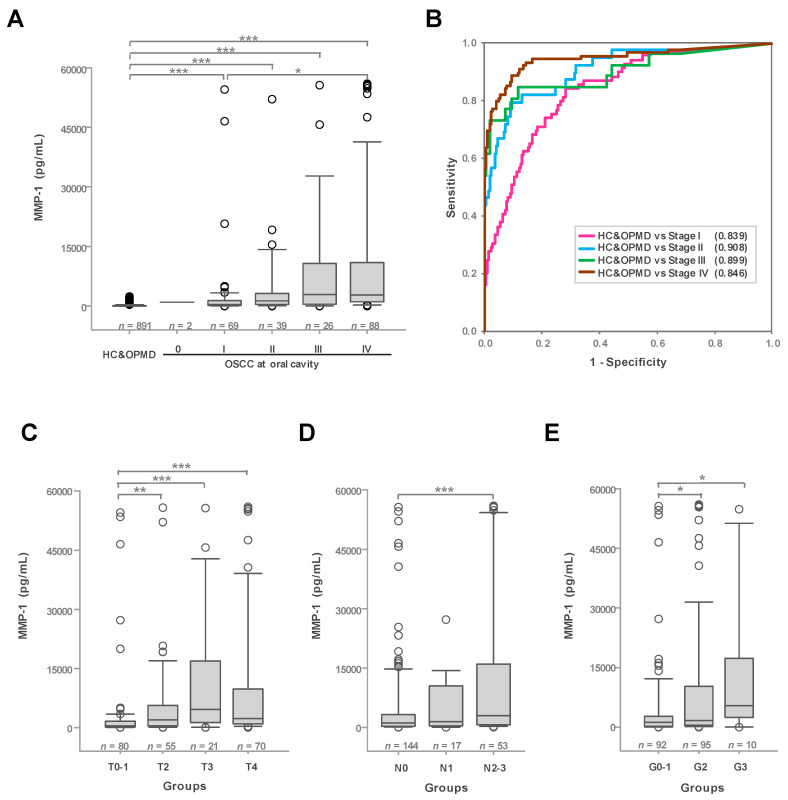
Differential expression levels of salivary MMP-1 in oral cavity cancer patients with advanced stages and grades. (**A**) Associations between salivary MMP-1 content in HC, OPMD, and different stages (0, I, II, III, and IV) of oral cavity cancer. * *p* < 0.05, ** *p* < 0.01, *** *p* < 0.001. (**B**) ROC curve showing the diagnostic efficacy of salivary MMP-1 for different stages (I–IV) of oral cavity cancer versus non-cancerous groups (HC and OPMD). (**C**–**E**) Associations between salivary MMP-1 content in oral cavity cancer specimens with tumor size (pT status, T0–T4) (**C**), neck lymph node metastasis (pN status, N0–N3) (**D**), and grade (G0–G3) (**E**).

**Table 1 cancers-12-02273-t001:** Demographics of clinical cases in this study.

Characteristics	Study Subjects (Case Number)				*p* Value (Chi-Square)
	HC	OPMD I	OPMD II	OSCC	
Total case number	313	305	273	269	-
Years of sampling 2011–2013/2014–2016	154/159	190/115	125/148	91/178	<0.001
Hospital of sampling CMMC/CGMH-KH	210/103	136/169	169/104	136/133	<0.001
Gender Male/Female	303/10	301/4	268/5	263/6	0.427
Range/Median age (years)	21–76/48	22–75/52	20–84/51	27–84/55	<0.001 ^a^
Individual habit					
Betel nut chewing (B) (No/Yes) Smoking (C) (No/Yes)	121/192 34/279	87/218 5/300	32/241 24/249	36/233 33/236	<0.001 <0.001
None/one/both of B and C	25/105/183	1/90/214	6/44/223	13/43/213	<0.001
Saliva appearance Clear/Yellow/Red	240/49/24	222/58/25	183/61/29	176/68/25	0.053

^a^ Difference in ages between groups was tested using the Kruskal–Wallis test.

**Table 2 cancers-12-02273-t002:** Summary of ROC curve analysis between non-cancerous and cancerous groups.

Non-Cancerous Group (Case No.)	Cancerous Group (Case No.)	Area Under ROC Curve (AUC)	95% CI of AUC	Best Cut-Off (Maximum of Youden’s Index)			
				Cut-Off Value (pg/mL)	Sensitivity	Specificity	Accuracy
HC (313)	OSCC (269)	0.898	0.873–0.924	207.6	78.07%	88.18%	88.18%
OPMD (578)		0.872	0.844–0.899	256.5	76.58%	85.29%	85.29%
HC and OPMD (891)		0.881	0.855–0.907	256.5	76.58%	86.76%	86.76%
HC and OPMD (891)	C02—Other and unspecified parts of tongue (64)	0.896	0.846–0.945	190.5	84.38%	83.28%	83.35%
	C03—Gum (37)	0.936	0.897–0.976	489.2	71.38%	92.59%	92.02%
	C06.0—Cheek mucosa (80)	0.900	0.857–0.942	256.9	83.75%	86.76%	86.51%
	Multiple sites at oral cavity (43)	0.877	0.817–0.937	381.3	74.42%	90.57%	89.83%
	Overall OSCCs at oral cavity (237)	0.899	0.875–0.924	256.5	79.32%	86.76%	85.20%
	Overall OSCCs at oropharynx (27)	0.738	0.621–0.856	122.5	70.37%	74.64%	74.51%
HC and OPMD (891)	Stage I (69)	0.839	0.791–0.886	97.5	84.06%	71.60%	72.50%
	Stage II (39)	0.908	0.854–0.962	397.2	79.49%	90.80%	90.32%
	Stage III (26)	0.899	0.815–0.982	300	84.62%	88.33%	88.22%
	Stage IV (88)	0.946	0.913–0.978	373.4	88.64%	90.46%	90.30%

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
