# Peer review of "Verification of Saliva Matrix Metalloproteinase-1 as a Strong Diagnostic Marker of Oral Cavity Cancer"

_cancers, 2020, doi:10.3390/cancers12082273_

Round 1

Reviewer 1 Report

This is a very interesting study that analyzes matrix metalloproteinase-1 (MMP-1) as a diagnostic biomarker in oral squamous cell carcinoma (OSCC). Authors consider that mass spectrometry-based quantitation method is not suitable for routine quantitative analysis of large numbers of samples in clinical practice, they produced specific antibodies against MMP-1 and developed an enzyme-linked immunosorbent assay (ELISA). To further verify the effectiveness of MMP-1 as a salivary biomarker for oral cancer detection, they used this in-house–developed ELISA to analyze levels of MMP-1 in 1160 clinical saliva samples from healthy subjects and patients with oral potentially malignant disorders (OPMD) or OSCC. The findings are really relevant, however, there are concerns needing to be addressed:

1. How patients were obtained: randomly, consecutively, other? In the material and methods section, it is necessary to explain it.

  1. Did the authors consider other risk factors such as alcohol use?
  2. How is the cutoff point decided? Youden's index/ Youden's J statistic in conjunction with ROC analysis is considered the best approach to obtain the cutoff point. This cutoff would be used to be a diagnostic test. Due to cutoff value is very important in this study could you use this approach?
  3. Would MMP-1 be increased in other cancer types? Would patients with inflammatory systemic or oral diseases exhibit similar changes? i.e MMP-1 is increased in periodontal disease. Periodontal disease is a confounding variable that was not taken into account in the present study. Would be possible to get this variable to include in the statistical analysis. If is not possible to get this information you have to include in the discussion section this important limitation of the study. In addition, you have to include references indicating this increase in periodontal diseases. Also, in conclusion, you should include that in the future confounding variables should be analyzed.
  4. The present study includes oropharynx and tonsil as OSCC. However, these anatomic sites are not considered “cancer of the oral cavity” DOI: 10.1016/j.oraloncology.2018.08.013;PMID: 30220324) (doi: 10.1007/s00784-020-03421-2; PMID: 32601998). Both anatomic locations should be removed from this study.
  5. Methodological data from the present study that were included in the discussion section should be removed (lines 305-308).

Reviewer 2 Report

The study is extremely interesting and useful. The topic is innovative, opening new scenarios for OSCC research and clinical management/diagnosis. Still, some key passages need to be clarified in order to allow pubblication. Here are my line per line suggestions

Line 114: Is this study prospective or retrospective? Author should clarify, and specify how patients were included (clarify inclusion creteria in the methods).

Ln 121: Why did you not use the latest version of AJCC (8th)? Patients should re-staged accordingly.

Ln 186-187: This sentence should be deleted.

Ln 201: Delete "in line with our expectation"; comments should be avoided in the "result"section.

Ln 209: On what biological bases did the authors divide the lesion into these groups? What is the significance of this further re-group? Significance per single site should be showed.

Ln 218: Author should treat more extensively the result of specificity and sensitivity for every OSCC and per site.

Discussion: do not recapitulate introduction, M&M or results in this section (e.g.: from line 303 on).

Ln 357: 1)Screening of high-risk population can be reached with a zero-cost white light evaluation performed by a trained clinician. Your results show great potential, but applicability needs to be evaluated with different types of prospective studies, evalueting all the cost-benefit factors. 2) and 3) Same as point 1, your test should to be essayed for effective useful application vs. actual gold standards (MRI, CT, biopsies...). It should be underlined that these are potential future applications, which needs to be properly tested with further studies.

Ln 365: Further studies should also show the benefit of your methodology when compared to a white-light inspection performed by a trained clinician. Furthermore, as you have partially stated, the efficacy of screening tests should always be evaluated considering epidemiology. 

Round 2

Reviewer 2 Report

Authors have followed the corrections partially. 

Ln 121: Latest version of AJCC classification should always be used, even when patients selection has been done over a long period of time. In case re-evaluation is not possible, whatever the reasons, authors should state that clearly, specifying that it should have been done but it could not have been done, and for which reason.

Ln 209: Oral mucosa can be divided into three main categories basing on histology:

-lining mucosa, nonkeratinized, stratified;

-masticatory mucosa, keratinized, stratified;

-specialized mucosa (e.g. taste buds).

Grouping two sites with different histology in name of "closeness" is thus a major mistake. For example, cheeck (lining) and trigon (masticatory), even if close, have different biology and different reaction to carcinogens (smoke, betel...), and thus cannot be grouped together in a molecular biology study. The biological differences of carcinogenesis of different types of epithelium are confirmed by prognostic and epidemiological data: floor of mouth cancer (lining mucosa) has a terrrible prognosis (also because of anatomical factor), while gum cancer is more rare. The same can be said about OPMD; a leukoplakia from the floor of mouth should worry more than a leukoplakia of the gum, confirming biological differences underlying. It would be different in a clinical study, in which this type of grouping could be partially justified, but would need different types of additional statistical analysis - and a bigger sample. 

Furthermore, grouping data to gain statistical significance only is a major bias per se. 

The article cannot be accepted in present form.

  1. Ten Cate's Oral Histology, Elsevier.
  2. WHO classification of Head & Neck Tumour, IARC. 

Round 3

Reviewer 2 Report

The authors have followed the suggestion. The paper is now suitable for publication.